# A digital twin reproducing gene regulatory network dynamics of early *Ciona* embryos indicates robust buffers in the network

**Miki Tokuoka[¤], Yutaka Satou** *

Department of Zoology, Graduate School of Science, Kyoto University, Sakyo, Kyoto, Japan

¤ Current address: Department of Advanced Medical Technologies, National Cerebral and Cardiovascular Center Research Institute (NCVC), Suita, Osaka, Japan

* yutaka@ascidian.zool.kyoto-u.ac.jp

**Data Availability Statement:** All data are included in the main document and supplemental materials. The computer code is deposited in Zenodo (doi: 10.5281/zenodo.7604201).

**Funding:** This research was supported by grants from the Japan Society for the Promotion of

## Abstract

How gene regulatory networks (GRNs) encode gene expression dynamics and how GRNs evolve are not well understood, although these problems have been studied extensively. We created a digital twin that accurately reproduces expression dynamics of 13 genes that initiate expression in 32-cell ascidian embryos. We first showed that gene expression patterns can be manipulated according to predictions by this digital model. Next, to simulate GRN rewiring, we changed regulatory functions that represented their regulatory mechanisms in the digital twin, and found that in 55 of 100 cases, removal of a single regulator from a conjunctive clause of Boolean functions did not theoretically alter qualitative expression patterns of these genes. In other words, we found that more than half the regulators gave theoretically redundant temporal or spatial information to target genes. We experimentally substantiated that the expression pattern of *Nodal* was maintained without one of these factors, Zfpm, by changing the upstream regulatory sequence of *Nodal*. Such robust buffers of regulatory mechanisms may provide a basis of enabling developmental system drift, or rewiring of GRNs without changing expression patterns of downstream genes, during evolution.

## Author summary

Although regulatory relationships in gene regulatory networks have been studied extensively during the past two decades, it is not fully understood how gene regulatory networks produce gene expression dynamics. One major reason is the difficulty of experimentally determining "regulatory functions" that mathematically describe how individual genes are regulated. Because of simplicity of embryonic structure and compactness of the genome, ascidian embryos provide an ideal system for analyzing dynamics of gene regulatory networks. In 32-cell ascidian embryos, there are only 13 regulatory genes that initiate expression. Regulatory functions for these genes, which are represented as Boolean functions, collectively constitute a digital twin that reproduces gene expression dynamics at single-cell resolution throughout the whole embryo. The digital twin is a

Science and CREST, Japan Science and Technology Agency (21H02486 and JPMJCR22N6 to YS, and 20J40280 and 22K06250 to MT). The funders had no role in study design, data collection and analysis, decision to publish, or preparation of the manuscript.

**Competing interests:** The authors have declared that no competing interests exist.

powerful tool for exploring mechanisms that are not readily accessible in experiments using real embryos. We found that over half the regulators in regulatory functions are potentially redundant for specifying temporal and spatial gene expression patterns. It is generally believed that regulatory functions have changed frequently during evolution, and our results show that a gene regulatory network in early ascidian embryos has robust buffers, which constitute a basis for developmental system drift.

## Introduction

Gene regulatory networks control gene expression dynamics during animal development. Although the structure of such networks, which represent connections among regulatory genes, has been studied extensively, it is not fully understood how these networks control gene expression dynamics. This is partly because it is laborious and challenging to mathematically represent how individual genes are regulated.

Ascidians are ideal chordates for studying gene regulatory networks because of their simple genome structure and simple embryonic structure. Indeed, the network structure of early embryonic stages has been analyzed extensively [1,2]. In particular, regulatory mechanisms in 32-cell embryos, in which germ layers are largely specified (Fig 1A), have been studied in detail [3–6]. A comprehensive expression assay [7] revealed that 13 regulatory genes begin to be expressed in 32-cell embryos. Regulatory mechanisms for 12 of these 13 genes have been described mathematically as Boolean functions, which represent necessary and sufficient conditions for their expression [5] (see an example for Boolean functions in Fig 2A). In other words, combinations of upstream regulatory factors that induce expression of these genes are represented as Boolean formulae, which we call regulatory functions (RFs). Expression patterns of these 12 genes can be predicted by calculations using these functions and activity of upstream factors. The RF for the remaining gene, *Nodal*, has also been determined provisionally. Necessary factors for *Nodal* expression were identified by our exhaustive examination of functions of regulatory genes expressed before the 32-cell stage. However, we did not always succeed in inducing *Nodal* expression in real embryos in conditions in which the tentative RF predicted *Nodal* activation [5]. This suggests that an additional modulator for transcription factors or signaling pathways may be involved in *Nodal* regulation. Determination of the *Nodal* RF will enable us to calculate expression patterns of all genes that begin to be expressed at the 32-cell stage. In other words, these RFs collectively constitute a digital twin, and we will be able to simulate gene expression dynamics in early ascidian embryos with this digital tool. In the present study, we first devised such a digital twin and then showed that gene expression patterns can indeed be manipulated in real embryos according to predictions from the digital twin. Using the digital twin, we also examined how changes in RFs potentially affected gene expression, because changes in RFs are thought to have occurred frequently during evolution but are difficult to assess experimentally using real embryos. We found robust buffers of regulatory mechanisms, which potentially permits rewiring of GRNs without changing expression patterns.

In the present study, RFs are represented as Boolean functions, as was successfully done in other systems [8–11]. There are several forms to represent a Boolean function, and disjunctive normal forms (DNFs) were used to represent RFs in a previous study [5] and in the present study, because this form is more easily interpreted from a biological viewpoint. DNFs are represented as the sum (disjunction; OR; ∨) of products (conjunction; AND; ∧) like A∧B∨A∧¬C (¬: negation, NOT). In this example, "A∧B" and "A∧¬C" are conjunctive clauses. If the RF for

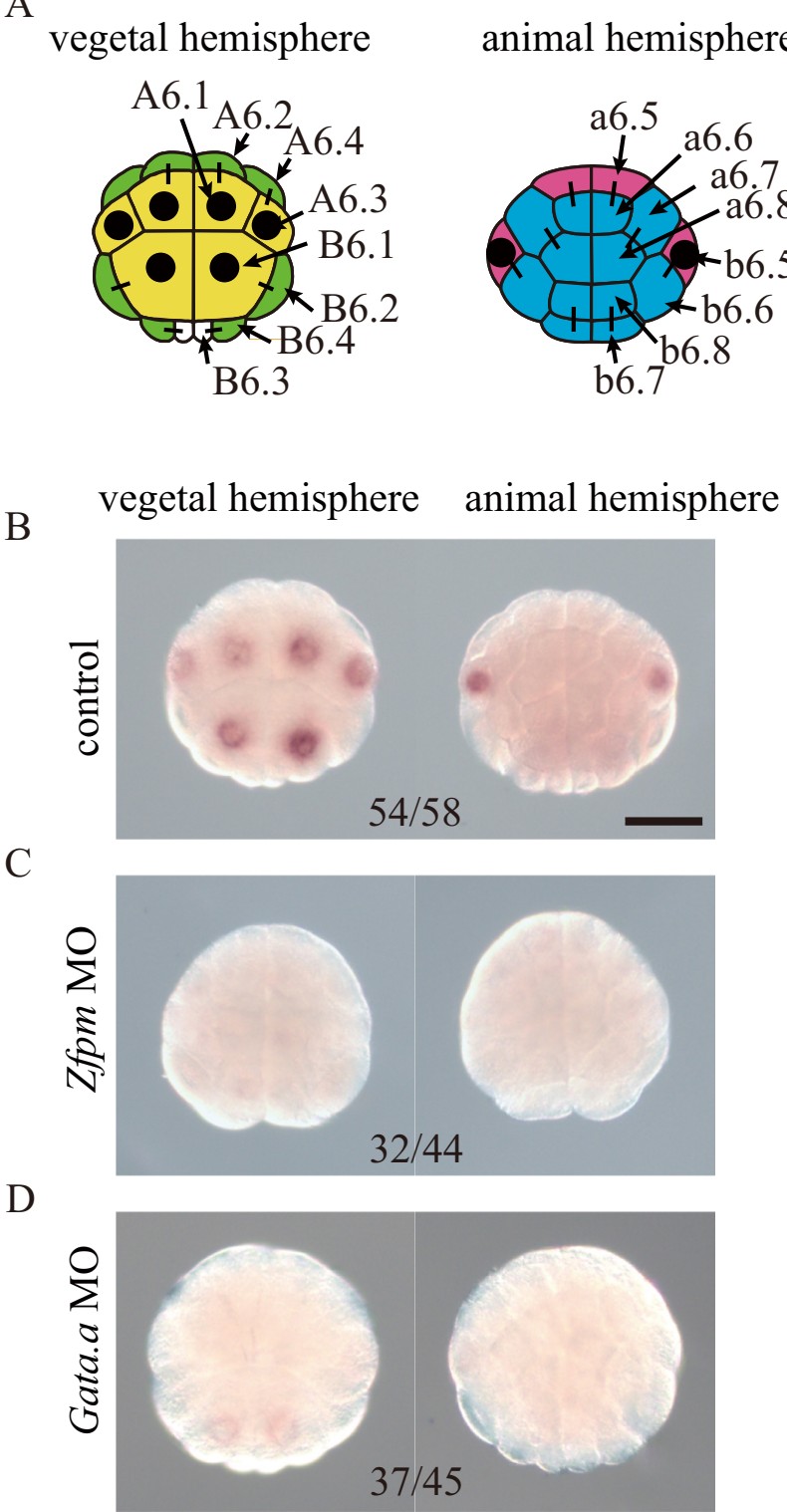

**Fig 1. *Nodal* is expressed under control of *Zfpm* in four pairs of cells in 32-cell ascidian embryos.** (A) Illustrations showing expression patterns of *Nodal* at the 32-cell stage. Cells with *Nodal* expression are marked by black dots. Yellow cells have endodermal fate. Green cells give rise to mesodermal tissues and nerve cord. Cyan cells give rise to epidermal cells, and magenta cells give rise to neural cells. Cell names of a bilaterally symmetrical embryo are shown on the right. (B–D) *In situ* hybridization revealed *Nodal* expression in normal embryos (B) and embryos injected with the *Zfpm* MO

(C) or *Gata.a* MO (D) at the 32-cell stage. Total numbers of embryos examined and numbers of embryos that photographs represent are shown within the panels. Scale bar, 50μm.


gene *X* is represented A∧B∨A∧¬C, this means that *X* is expressed when the upstream factors, A and B, are both present or when the upstream factor A is present and the upstream factor C is absent (or both). RFs can also be expressed as a form of truth tables. A truth table

A

$F^{Nodal}$= Foxa.a∧Fgf9/16/20∧β-catenin∧Gata.a∧Zfpm ∨ (∧: AND, ∨: OR, ¬: NOT)
Tbx6-r.b∧β-catenin∧Gata.a∧Zfpm ∨
Sox1/2/3∧¬Foxa.a∧¬Foxd∧Fgf9/16/20∧¬Efna.d∧¬β-catenin∧Gata.a∧Zfpm ∨
Fgf9/16/20∧¬Gdf1/3-r∧¬Admp∧¬Prdm1-r∧¬Foxa.a∧¬β-catenin∧Gata.a

| | | | Upstream regulaotry factors | | | | | | | | | | | Calculation | | | | | Experimental results |
|---|---|---|---|---|---|---|---|---|---|---|---|---|---|---|---|---|---|---|---|
| | Experimental conditions | Cells | Foxa.a | Foxd | Prdm1-r | Sox1/2/3 | Tbx6-r.b | Admp | Efna.d | Fgf9/16/20 | Gdf1/3-r | β-catenin | Zfpm | Gata.a | conj. clause 1 | conj. clause 2 | conj. clause 3 | conj. clause 4 | $F^{Nodal}$ | vegetal hemisphere / animal hemisphere |
| B | normal | Vegetal | 0 | 0 | 0 | 0 | 0 | 0 | 0 | 0 | 0 | 1 | 0 | 1 | 0 | 0 | 0 | 0 | 0 | see Figure 3D |
| | | Animal | 0 | 0 | 0 | 0 | 0 | 0 | 0 | 0 | 0 | 0 | 0 | 1 | 0 | 0 | 0 | 0 | 0 | |
| C | FGF treatment | Vegetal | 0 | 0 | 0 | 0 | 0 | 0 | 0 | 1 | 0 | 1 | 0 | 1 | 0 | 0 | 0 | 0 | 0 | 72/79 (91%) |
| | | Animal | 0 | 0 | 0 | 0 | 0 | 0 | 0 | 1 | 0 | 0 | 0 | 1 | 0 | 0 | 0 | 1 | 1 | |
| D | FGF treatment+ *Foxa.a* mRNA +*Zfpm* mRNA | Vegetal | 1 | 0 | 0 | 0 | 0 | 0 | 0 | 1 | 0 | 1 | 1 | 1 | 1 | 0 | 0 | 0 | 1 | 65/72 (90%) |
| | | Animal | 1 | 0 | 0 | 0 | 0 | 0 | 0 | 1 | 0 | 0 | 1 | 1 | 0 | 0 | 0 | 0 | 0 | |
| E | *Tbx6-r.b* mRNA | Vegetal | 0 | 0 | 0 | 0 | 1 | 0 | 0 | 0 | 0 | 1 | 0 | 1 | 0 | 0 | 0 | 0 | 0 | 113/113 (100%) |
| | | Animal | 0 | 0 | 0 | 0 | 1 | 0 | 0 | 0 | 0 | 0 | 0 | 1 | 0 | 0 | 0 | 0 | 0 | |
| F | T*bx6-r.b* mRNA+*Zfpm* mRNA | Vegetal | 0 | 0 | 0 | 0 | 1 | 0 | 0 | 0 | 0 | 1 | 1 | 1 | 0 | 1 | 0 | 0 | 1 | 63/81 (78%) |
| | | Animal | 0 | 0 | 0 | 0 | 1 | 0 | 0 | 0 | 0 | 0 | 1 | 1 | 0 | 0 | 0 | 0 | 0 | |
| G | FGF treatment+*Sox1/2/3* mRNA +*Prdm1-r* mRNA | Vegetal | 0 | 0 | 1 | 1 | 0 | 0 | 0 | 1 | 0 | 1 | 0 | 1 | 0 | 0 | 0 | 0 | 0 | 40/43 (93%) |
| | | Animal | 0 | 0 | 1 | 1 | 0 | 0 | 0 | 1 | 0 | 0 | 0 | 1 | 0 | 0 | 0 | 0 | 0 | |
| H | FGF treatment+*Sox1/2/3* mRNA +*Prdm1-r* mRNA+*Zfpm* mRNA | Vegetal | 0 | 0 | 1 | 1 | 0 | 0 | 0 | 1 | 0 | 1 | 1 | 1 | 0 | 0 | 0 | 0 | 0 | 34/48 (71%) |
| | | Animal | 0 | 0 | 1 | 1 | 0 | 0 | 0 | 1 | 0 | 0 | 1 | 1 | 0 | 0 | 1 | 0 | 1 | |

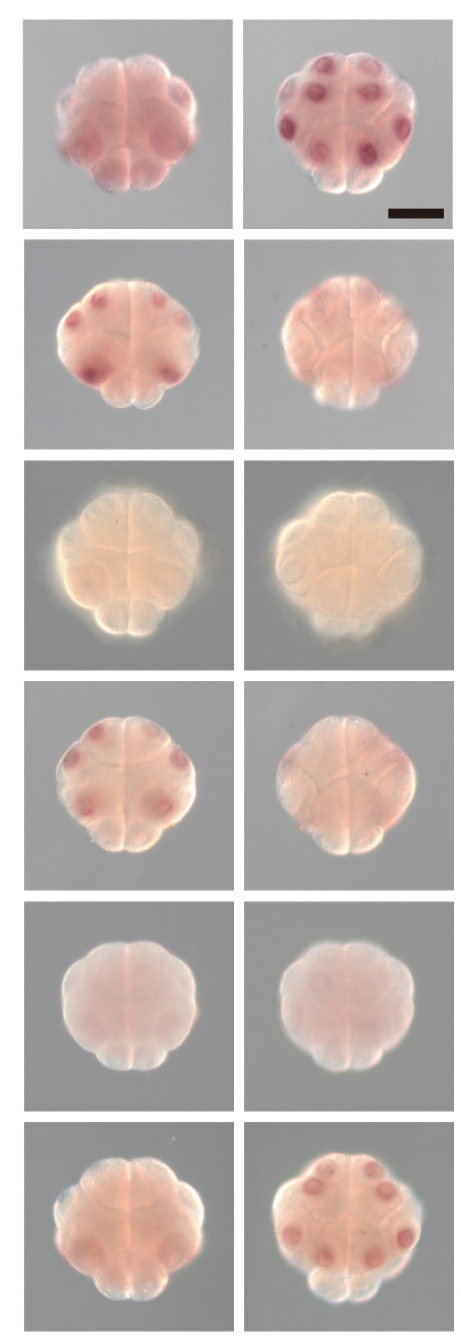

**Fig 2. Prediction and experimental validation of *Nodal* expression in experimental conditions using 16-cell embryos.** (A) The RF of *Nodal* consists of four conjunctive clauses. Note that the first three conjunctive clauses are responsible for expression in A6.1/A6.3/B6.1, B6.1, and b6.5 of normal embryos, respectively, while the fourth conjunctive clause is responsible for expression in conditions that do not appear in normal embryos. (B–H) Distributions of upstream factors in normal and experimental conditions are shown on the left. This distribution pattern was inferred on the basis of observations in previous studies [6,13–24]. Differences from the pattern of normal 16-cell embryos are indicated with magenta. The next four columns show whether each conjunctive clause is satisfied (1) or unsatisfied (0). The last column indicates whether the RF is satisfied (1) or unsatisfied (0). On the right, *Nodal* expression in real embryos revealed by *in situ* hybridization is shown. Total numbers of embryos examined and numbers of embryos that photographs represent are shown on the right. An experimental condition shown in (A) was examined in experiments shown in Fig 3D; therefore, no photographs are included in this panel. Note that photographs in (C) are the same as those in S4A Fig. Scale bar, 50 μm.

comprehensively represents expression of a target gene in various combinations of upstream factors. In the present study, we consider 18 upstream factors (see S1 Table); therefore, there are 262,144 (= $2^{18}$) possible combinations. It is not realistic to comprehensively examine all these conditions experimentally. For this reason, in our previous study [5], among DNFs that are compatible with gene expression patterns in normal embryos and a limited number of experimental embryos (i.e. partially filled truth tables), DNFs with the smallest number of conjunctive clauses and the smallest number of literals (upstream regulators) were considered as primary candidates; if multiple candidates were obtained, we repeated experiments until we obtained a unique candidate DNF. In other words, candidate DNFs were determined under the assumption that the simplest DNF (DNF with the smallest number of conjunctive clauses and the smallest number of upstream regulators) that explains all observations is most likely. In experiments for determining DNFs, we used embryos injected with morpholino antisense oligonucleotides (MO) for knockdown and/or treated with an inhibitor for the MAPK signaling pathway. Then we experimentally verified whether candidate RFs correctly predicted gene expression under conditions that were not previously examined. In verification experiments, we used MOs, synthetic mRNAs, the MAPK signaling inhibitor, and a recombinant FGF protein. In this way, we determined RFs using experiments and theoretical analyses. The finding that RFs are successfully represented as Boolean functions indicates that qualitative, but not quantitative, control is important for gene expression in early ascidian embryos [5].

To predict gene expression patterns with these RFs, distribution patterns of the 18 upstream factors are necessary. Distribution of these upstream factors in normal embryos is based on observations in previous studies [6,12–24]. We assumed that descendants of cells expressing a transcription factor gene at the 16-cell stage express the encoded protein at the 32-cell stage because of a delay between gene expression and protein translation [5]. In the present study, we employed the same approach to determine the function of *Nodal*.

## Results and discussion

### Regulatory function of *Nodal*

Expression patterns predicted by the presumptive *Nodal* RF in various conditions are reproduced experimentally at the 32-cell stage [5]. If this *Nodal* RF represents sufficient conditions, *Nodal* expression could be induced in experimentally manipulated 16-cell embryos. However, we failed to induce *Nodal* expression in 16-cell experimental embryos, in which we reproduced conditions that were predicted to be sufficient to induce *Nodal* expression [5]. Therefore, it is likely that there is a missing activator that acts at the 32-cell stage, but not at the 16-cell stage. Because there is a delay between mRNA transcription and protein activity, we hypothesized that the mRNA encoding this missing factor should be expressed at the 16-cell stage. Since we have comprehensively identified genes encoding transcription factors and signaling molecules and have examined their functions [5,7], it is unlikely that this missing factor is a transcription factor or a signaling molecule.

*Zfpm*, a gene for a FOG family zinc finger protein, satisfies most of these criteria. It begins to be expressed in 16-cell embryos [21], and the encoded protein acts as a co-factor of Gata transcription factors [25]. Because *Nodal* is expressed in both the animal and vegetal hemispheres, the missing factor should also be expressed in both hemispheres. However, a previous study reported that expression of *Zfpm* was observed only in the animal hemisphere [21]. Therefore, we re-examined the expression pattern of *Zfpm* and found that it is, in fact, expressed in both hemispheres, except for the posterior-most germ-line cells at the 16-cell stage (S1A–S1C Fig) [Note that because transcription is repressed in germ-line cells [15,22], we did not consider germ-line cells in the present study]. Indeed, knockdown of *Zfpm* using a

MO and CRISPR-based knockout of *Zfpm* abolished *Nodal* expression at the 32-cell stage (Figs 1B and 1C, and S2). Because Zfpm is a co-factor of Gata [25], we confirmed that Gata.a is necessary for activation of *Nodal* by knockdown of *Gata.a*, using a specific MO (Fig 1D) [this MO has been used repeatedly in studies by different groups [14,17,21,26,27]; therefore, we did not further evaluate specificity of this MO in the present study]. Thus, *Zfpm* and *Gata.a* positively regulate *Nodal* at the 32-cell stage.

The tentative Boolean function for *Nodal*, which was determined previously [5], is represented in a DNF, and consists of four conjunctive clauses (S3 Fig). Because the first three conjunctive clauses, respectively, represent expression in A6.1/A6.3/B6.1, B6.1, and b6.5 cells of normal embryos, and because expression in these cells was abolished in *Zfpm* morphants (Fig 1C), it is likely that *Zfpm* is involved in expression represented by these three conjunctive clauses. On the other hand, the fourth conjunctive clause does not represent expression in normal embryos but represents expression in experimental conditions, e.g. expression in embryos in which *Gdf1/3-r* and *Admp* are simultaneously knocked down [5], and involvement of *Zfpm* in the fourth conjunctive clause was unclear from the above experiments. To test the fourth function, we used 16-cell embryos, which do not normally express *Nodal*. In 16-cell embryos, we created an experimental condition that satisfied the fourth conjunctive clause by incubating embryos in sea water containing FGF2, which mimicked overexpression of *Fgf9/16/20*. In such embryos, *Nodal* was expressed in the animal hemisphere (S4A Fig). The observation that FGF treatment induced *Nodal* expression in 16-cell embryos suggested that *Zfpm* may be unnecessary for the fourth conjunctive clause, because *Zfpm* begins to be expressed at the 16-cell stage; therefore it is unlikely that sufficient Zfpm protein is available at this stage. Indeed, injection of the *Zfpm* MO did not affect *Nodal* expression in 16-cell embryos incubated in sea water containing FGF2 (S4B Fig).

There was some ambiguity in the fourth conjunctive clause of the previous tentative RF for *Nodal*. Because the fourth conjunctive clause does not represent expression in normal embryos, this uncertainty was not resolved in the previous study [5]. In the present study, to resolve this enigma, we performed additional experiments. First, we injected an MO against *Gata.a* or *β-catenin* in addition to FGF2 treatment, and found that Gata.a and β-catenin regulate *Nodal* positively and negatively, respectively, in 16-cell embryos treated with FGF2 (S4C and S4D Fig). Second, because it has been suggested that *Prdm1-r* may negatively regulate *Nodal* in the fourth conjunctive clause [5], we injected *Prdm1-r* mRNA and treated injected embryos with FGF2, confirming that *Nodal* expression was lost (S4E Fig). Similarly, a previous study showed that embryos injected with *Foxa.a* mRNA and treated with FGF2 do not express *Nodal* at the 16-cell stage [5] (S4F Fig). These data indicate that both *Prdm1-r* and *Foxa.a* act as negative regulators of *Nodal*.

Using a method that we previously developed [5] and experimental results for *Nodal* expression in various conditions, which we described above and in our previous study [5] (S1 Table), we obtained the simplest DNF of the Boolean function for *Nodal* (Fig 2A). For this calculation, we assumed that descendants of cells expressing a transcription factor gene at the 16-cell stage would express the encoded protein at the 32-cell stage (S1 Table), and used the distribution pattern of upstream factors inferred from observations in previous studies [6,13–24].

We confirmed that *Nodal* expression can be induced in 16-cell embryos by various experimental manipulations in patterns that each of the conjunctive clauses of the *Nodal* RF predicted (Fig 2B–2H). That is, the RF predicted no expression of *Nodal* in normal 16-cell embryos (Fig 2B), and indeed *Nodal* is not expressed [7] (see also Fig 3D). Similarly, as we described above, 16-cell embryos incubated in sea water containing FGF2 expressed *Nodal* in the animal hemisphere, and this condition satisfies the fourth conjunctive clause of the RF

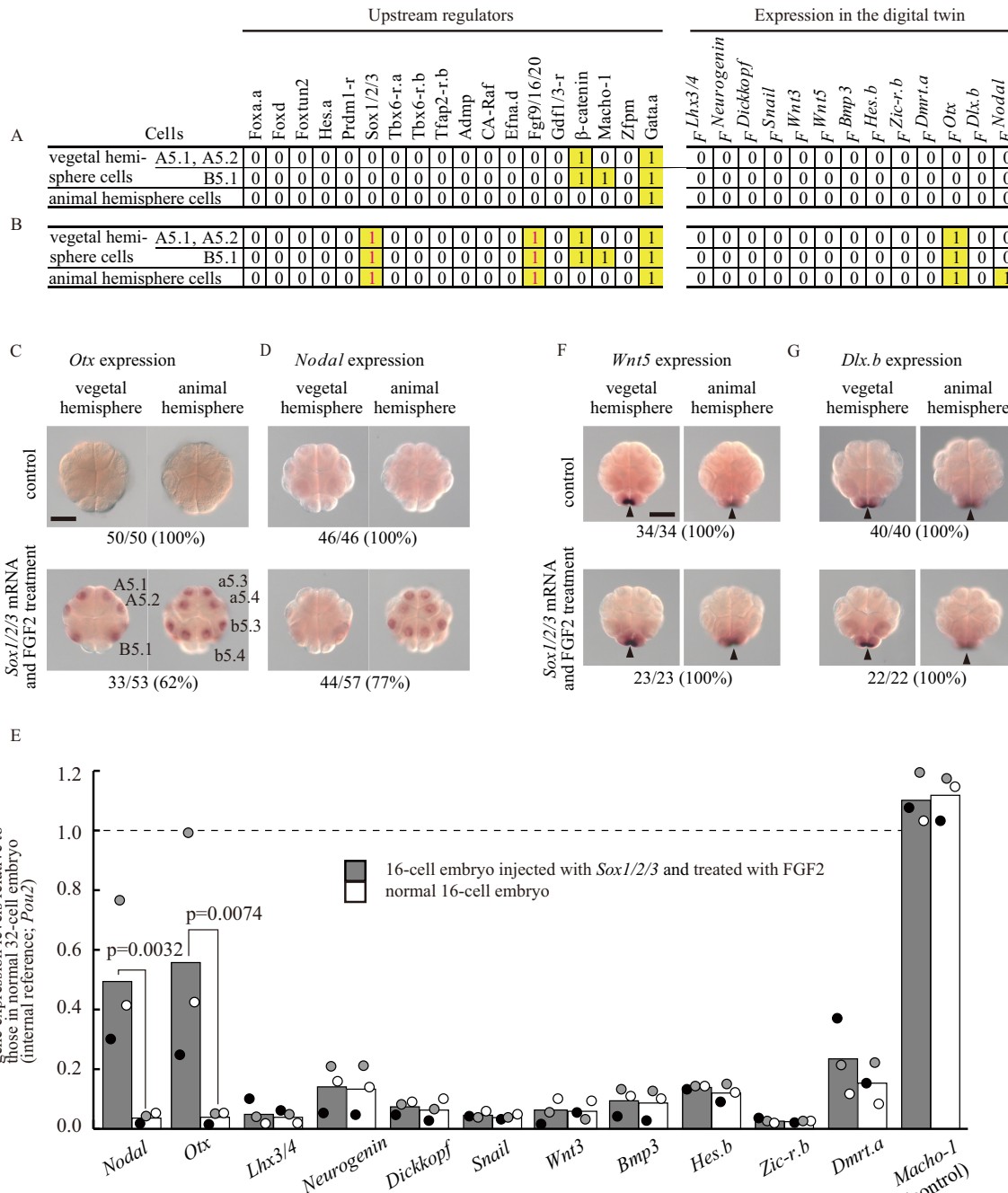

**Fig 3. Reproduction of the regulatory state of b6.5 of a 32-cell embryo in a digital twin and a real embryo at the 16-cell stage.** (A) The distribution of upstream regulatory factors in the normal 16-cell embryo (left; 1, presence; 0, absence), inferred on the basis of observations in previous studies [6,13–24], and calculated expression patterns of 13 genes that normally begin to be expressed at the 32-cell stage, in the digital twin at the 16-cell stage (right; 1, expression; 0, no expression). (B) The digital twin predicts that *Otx* and *Nodal*, but not the others, are expressed at the 16-cell stage, if Sox1/2/3 and Fgf9/16/20 act in all cells at the 16-cell stage (magenta). (C, D) Expression of (C) *Otx* and (D) *Nodal* was reproduced in real 16-cell embryos injected with *Sox1/2/3* mRNA and treated with FGF2, which mimicked overexpression of *Fgf9/16/20*, as the digital twin predicted, while unperturbed control 16-cell embryos do not express *Otx* or *Nodal*. Total numbers of embryos examined and numbers of embryos that photographs represent are shown. (E) Expression levels of the 11 genes, except *Wnt5* and *Dlx.b*, were examined using reverse transcription and quantitative PCR (RT-qPCR). Maternal *Pou2* was used as an internal reference, and another maternally expressed gene, *Macho-1*, was used as a control. Gene expression levels are shown as values relative to those in normal 32-cell embryos. Results from independent experiments using different batches of embryos are indicated by black, gray, and white dots, and averages are shown by bars. Delta Ct values were used for Student's t-tests, which showed significant differences in expression of *Nodal* and *Otx*, but not of the other genes (p>0.05). P-values for *Nodal* and *Otx*

are shown in the graph. (F, G) Because maternal mRNAs of *Wnt5* and *Dlx.b* complicate RT-qPCR measurements, expression of *Wnt5* and *Dlx.b* was examined by *in situ* hybridization. Arrowheads indicate maternal mRNAs localized to the posterior pole. No zygotic expression was evident in normal or experimental embryos. Total numbers of embryos examined and numbers of embryos that photographs represent are shown below the panels. Scale bars, 50 μm.

only in the animal hemisphere (Fig 2C). As we have already shown in S4 Fig, if we inject *Foxa.a* mRNA into unfertilized eggs and incubate them in sea water containing FGF2 after fertilization, *Nodal* is not expressed (this condition does not satisfy any conjunctive clauses). Then, we injected *Foxa.a* and *Zfpm* mRNAs into unfertilized eggs and incubated them in sea water containing FGF2. In this condition, the first conjunctive clause was satisfied in the vegetal hemisphere. That is, the RF predicted *Nodal* expression in the vegetal hemisphere, except for germ line cells, and such embryos indeed expressed *Nodal* in the vegetal hemisphere (Fig 2D). Similarly, we induced *Nodal* expression in 16-cell embryos according to the second and third conjunctive clauses by injecting mRNAs for *Tbx6-r.b* and *Zfpm*, and injecting mRNAs for *Sox1/2/3*, *Prdm1-r*, and *Zfpm* followed by FGF treatment, respectively (Fig 2E–2H). Thus, *Nodal* was expressed or not expressed in these experimental conditions, as the *Nodal* RF predicted.

As in the previous study [5], we assumed that the *Nodal* RF can be represented as a Boolean function. Our success suggests that this assumption is appropriate, and that qualitative controls are important for regulation in early *Ciona* embryos. The first three conjunctive clauses of the *Nodal* RF represent expression in A6.1/A6.3/B6.1, B6.1, and b6.5 of normal 32-cell embryos, respectively. On the other hand, the fourth conjunctive clause is not necessary for expression in normal embryos at the 32-cell stage.

## Reproduction of the pattern of a cell with neural fate of a 32-cell embryo at the 16-cell stage

The *Nodal* RF and RFs for the other 12 genes, which have been determined previously [5], collectively constitute a digital twin to model gene expression dynamics of the whole 32-cell ascidian embryo. An HTML-based program that implements these RFs enables us to calculate at single-cell resolution how the 13 regulatory genes are expressed under various conditions (http://ghost.zool.kyoto-u.ac.jp/sim32v2/). Because these RFs represent conditions necessary and sufficient for gene expression, the RFs allow us to predict expression patterns of these 13 genes even at the 16-cell stage. Therefore, at the 16-cell stage, we tried to reproduce the pattern that is normally seen in a cell pair (b6.5) of 32-cell embryos using the RFs, as b6.5 cells are important for patterning the neural plate at later stages [1,28,29]. In other words, because only *Nodal* and *Otx* are expressed in b6.5 cells of normal 32-cell embryos among the above 13 genes [7], we computationally tested various conditions and found a condition to specifically induce *Nodal* and *Otx* expression at the 16-cell stage. The digital twin predicted that *Nodal* and *Otx*, but not the other 11 genes, would be expressed in the animal hemisphere of 16-cell embryos with overexpression of Sox1/2/3 and Fgf9/16/20 (Fig 3A and 3B). *Otx* and *Nodal* were indeed expressed in 16-cell embryos injected with *Sox1/2/3* mRNA and treated with FGF2 (Fig 3C and 3D). We also confirmed that the remaining genes were not expressed or upregulated at the 16-cell stage in such experimental embryos. Expression levels of nine genes (excluding *Wnt5* and *Dlx.b*), as well as expression levels of *Nodal* and *Otx*, were measured by reverse transcription and quantitative PCR (RT-qPCR) (Fig 3E). Meanwhile, *Wnt5* and *Dlx.b* are maternally expressed, and these maternal mRNAs disturbed accurate measurement of zygotic expression by RT-qPCR. Therefore, these two genes were examined by *in situ* hybridization. Because their maternal mRNAs are localized at the posterior pole, these maternal mRNAs are easily discernible and no zygotic expression was observed in experimental embryos (Fig 3F

and 3G). These analyses showed that *Otx* and *Nodal*, but not the other 11 genes, are activated in 16-cell embryos when injected with *Sox1/2/3* mRNA and treated with FGF2. Thus, the expression pattern of b6.5 of 32-cell embryos was reproduced in the animal hemisphere of 16-cell embryos, as the digital twin predicted. In other words, the digital twin successfully reproduces gene expression dynamics of real embryos even under conditions that were not used for constructing the RFs.

In this way, we succeeded in reproducing the pattern of b6.5 of 32-cell embryos in 16-cell embryos with the aid of the digital twin. However, our success may not necessarily mean that these manipulated cells differentiate in the same way as normal b6.5 cells. We injected *Sox1/2/3* mRNA into eggs, and it probably persists after the stage at which endogenous *Sox1/2/3* mRNA diminishes. In addition, we did not examine how such manipulation changed expression of genes that begin to be expressed at the 16-cell stage. Therefore, it is not easy to predict the cell types into which these manipulated cells finally differentiate. Producing digital twins for 16-cell, 64-cell, and later embryos will resolve this problem, and a technique that degrades introduced mRNAs in a timely manner will also be useful for manipulating terminally differentiated cell types.

## Robust buffers of regulatory mechanisms revealed by the digital twin

Because RFs are primarily encoded in *cis*-regulatory regions, often in a complex and redundant manner [30–34], it is generally difficult to change regulatory mechanisms or RFs *in vivo*. Therefore, the digital twin provides a unique opportunity to examine effects of changes in RFs. In other words, the digital twin can predict how gene expression patterns are altered by changing RFs. In RFs for the 13 genes (S2 Table), 16 upstream regulators appear 100 times in total. Each of these 16 regulators appears multiple times in different conjunctive clauses (Fig 4). We found 55 cases in which removal of a single regulator from a single conjunctive clause of RFs did not qualitatively change expression patterns in normal 16-cell or 32-cell embryos (Fig 4). For example, *Lhx3/4*, *Neurogenin*, and *Dickkopf* are expressed in the same cells, and their RFs are commonly represented as Foxd∧Fgf9/16/20∧β-catenin (Fig 4). Our analysis using the digital twin shows that Fgf9/16/20∧β-catenin and Foxd∧β-catenin can establish the same expression pattern (S5A Fig). That is, even if regulatory circuits are rewired in the future and RFs for *Lhx3/4*, *Neurogenin*, and *Dickkopf* are changed to Fgf9/16/20∧β-catenin or Foxd∧β-catenin, the original gene expression pattern could be retained theoretically. It is noteworthy that this rewiring may include not only loss of binding sites for Foxd or Ets1/2 (an effector of the FGF pathway) but also reorganization of their *cis*-regulatory regions. This finding suggests that the observed theoretical redundancy may facilitate developmental system drift, or rewiring of GRNs without changing expression patterns of downstream genes. Note that the word "redundancy" here does not mean that redundant factors are not necessary for gene expression. Instead, it means that gene expression will not be changed without a redundant factor if regulatory regions are properly re-designed. In other words, information sufficient for specific gene expression is given to a target gene without a redundant factor. The digital twin also indicated that such rewiring would enable these genes to behave differently if expression patterns of upstream factors were changed (S5B, S5C, and S5D Fig). Conversely, it is theoretically possible that developmental system drift created this redundancy. However, this latter possibility is unlikely at least in cases of *Lhx3/4*, *Neurogenin*, and *Dickkopf*. If so, it would indicate that developmental system drift changed regulatory mechanisms of these genes independently three times, so that these three genes became controlled by the same combination of regulatory factors. Thus, the above results indicate that more than half the regulators give theoretically redundant temporal or spatial information to target genes, or that some do not furnish such information.

$$F^{Lhx3/4} = \text{Foxd} \wedge \text{Fgf9/16/20} \wedge \beta\text{-catenin}$$

$$F^{Neurogenin} = \text{Foxd} \wedge \text{Fgf9/16/20} \wedge \beta\text{-catenin}$$

$$F^{Dickkopf} = \text{Foxd} \wedge \text{Fgf9/16/20} \wedge \beta\text{-catenin}$$

$$F^{Snail} = \text{CA-Raf} \wedge \text{Macho1} \vee$$
$$\text{Tbx6-r.b}$$

$$F^{Wnt3} = \text{CA-Raf} \wedge \text{Macho1} \vee$$
$$\text{Tbx6-r.b}$$

$$F^{Wnt5} = \text{CA-Raf} \wedge \text{Macho1} \vee$$
$$\text{Tbx6-r.b}$$

$$F^{Bmp3} = \text{Foxa.a} \wedge \text{Foxd}$$

$$F^{Hes.b} = \text{Foxd} \wedge \text{Fgf9/16/20} \vee$$
$$\neg \text{Sox1/2/3} \wedge \neg \text{Hes.a} \wedge \text{Fgf9/16/20} \wedge \neg \text{Efna.d} \wedge \neg \text{Prdm1-r}$$

$$F^{Zic-r.b} = \text{Foxa.a} \wedge \text{Foxd} \wedge \text{Fgf9/16/20} \wedge \neg \beta\text{-catenin} \wedge \text{Gata.a} \vee$$
$$\text{Macho1} \wedge \neg \beta\text{-catenin} \wedge \text{Gata.a} \vee$$
$$\neg \text{Sox1/2/3} \wedge \neg \text{Hes.a} \wedge \text{Fgf9/16/20} \wedge \neg \beta\text{-catenin} \wedge \text{Gata.a} \vee$$
$$\text{Foxa.a} \wedge \neg \text{Hes.a} \wedge \text{Fgf9/16/20} \wedge \neg \beta\text{-catenin} \wedge \text{Gata.a}$$

$$F^{Dmrt.a} = \text{Sox1/2/3} \wedge \text{Foxa.a} \wedge \neg \text{Foxd} \wedge \text{Fgf9/16/20} \wedge \neg \text{Efna.d} \wedge \neg \beta\text{-catenin} \wedge \text{Gata.a}$$

$$F^{Otx} = \text{Macho1} \wedge \neg \beta\text{-catenin} \wedge \text{Gata.a} \vee$$
$$\text{Tbx6-r.a} \wedge \text{Fgf9/16/20} \vee$$
$$\text{Tbx6-r.b} \wedge \text{Fgf9/16/20} \vee$$
$$\neg \text{Foxd} \wedge \text{Fgf9/16/20} \wedge \neg \text{Efna.d} \wedge \text{Gata.a} \vee$$
$$\text{Fgf9/16/20} \wedge \neg \text{Gdf1/3-r} \wedge \neg \text{Admp}$$

$$F^{Dlx.b} = \text{Sox1/2/3} \wedge \text{Foxa.a} \wedge \neg \text{Foxd} \wedge \neg \beta\text{-catenin} \wedge \text{Gata.a} \vee$$
$$\text{Sox1/2/3} \wedge \text{Efna.d} \vee$$
$$\text{Sox1/2/3} \wedge \neg \text{Foxd} \wedge \neg \text{Fgf9/16/20}$$

$$F^{Nodal} = \text{Foxa.a} \wedge \text{Fgf9/16/20} \wedge \beta\text{-catenin} \wedge \text{Gata.a} \wedge \text{Zfpm} \vee$$
$$\text{Tbx6-r.b} \wedge \beta\text{-catenin} \wedge \text{Gata.a} \wedge \text{Zfpm} \vee$$
$$\text{Sox1/2/3} \wedge \neg \text{Foxa.a} \wedge \neg \text{Foxd} \wedge \text{Fgf9/16/20} \wedge \neg \text{Efna.d} \wedge \neg \beta\text{-catenin} \wedge \text{Gata.a} \wedge \text{Zfpm} \vee$$
$$\text{Fgf9/16/20} \wedge \neg \text{Gdf1/3-r} \wedge \neg \text{Admp} \wedge \neg \text{Prdm1-r} \wedge \neg \text{Foxa.a} \wedge \neg \beta\text{-catenin} \wedge \text{Gata.a}$$

**Fig 4. Theoretically redundant or dispensable regulation for establishing specific expression patterns in RFs of the 13 genes that initiate expression at the 32-cell stage.** In RFs for the 13 genes, magenta indicates 55 cases of regulators that give theoretically redundant temporal or spatial information to target genes. For example, the RF for *Lhx3/4* is Foxd∧Fgf9/16/20∧β-catenin, while Foxd∧β-catenin and Fgf9/16/20∧β-catenin can establish the same expression pattern. Note that we did not perform tests in which two or more factors were simultaneously removed; therefore, the above presentation did not necessarily mean that *Lhx3* expression pattern could be reproduced by β-catenin alone. Meanwhile, because Efna.d antagonizes signaling by Fgf9/16/20 [19,20,47,48], the term "¬Efna.d" implies that Fgf9/16/20 acts. Similarly, because β-catenin antagonizes Gata.a activity [17], the term "¬β-catenin" implies that Gata.a acts. Therefore, although removal of Gata.a or Fgf9/16/20, shown in cyan from the digital twin, could not change expression patterns theoretically, these predictions would not be reproduced in real embryos. Note that three upstream regulatory factors, Ets1/2, Tcf7, and Pem-1, are not included, but they are implicitly considered in the RFs: (1) Ets1/2 acts as an effector of the MAPK pathway, which is regulated by Fgf9/16/20, constitutively active (CA) Raf and Efna.d [6,26,47]; (2) Tcf7 acts as a positive regulator with nuclear β-catenin; (3) Pem-1 represses transcription by regulating the function of RNA polymerase II in the germ-line cells [15,22], which are not considered in the present study.

It is unlikely that the observed redundancy is an artifact derived from our method to determine RFs. First, we identified upstream regulatory factors in an unbiased way as we mentioned in the Introduction section. Second, we started with considering all theoretically possible conjunctive clauses to determine RFs, as we explained in detail in the Introduction section; therefore, it is unlikely that this method favors or disfavors redundancy of the network. Third, RFs represent necessary and sufficient conditions for expression of individual target genes, and their sufficiency was experimentally verified [5] (see also Fig 2).

Although more than half the regulators give theoretically redundant temporal or spatial information to target genes, they are necessary for expression of their target genes in real embryos. The above computational analysis showed that Foxd and Fgf9/16/20 theoretically gave redundant cues for regulation of *Lhx3/4*, whereas *Lhx3* expression is lost in embryos injected with an MO against *Foxd* or *Fgf9/16/20* [5]. Similarly, the present and previous studies have experimentally shown that each of the observed redundant factors act to establish specific expression patterns of the 13 target genes [4,5,14,19,20,26,35]. Exceptions include involvement of β-catenin and Gata.a in regulation of *Dmrt.a* and *Dlx.b*. Therefore, we confirmed that these factors indeed regulate expression of *Dmrt.a* and *Dlx.b* (S6 Fig). Thus, redundant temporal or spatial information is given to the 13 target genes.

In this way, the above hypothetical experiment using the digital twin indicates that wide tolerance in the gene regulatory network may constitute evolutionary potential to rewire networks, and may have contributed to evolution of gene regulatory networks. Thus, the digital twin makes it possible to analyze mechanisms that are difficult to access with experiments using real embryos.

## Rewiring of the regulatory mechanism for *Nodal*

As shown in Fig 4, Zfpm is a factor that does not furnish spatial information for specific expression of *Nodal*. Even if Zfpm is removed simultaneously from all three conjunctive clauses that contain Zfpm, *Nodal* expression is theoretically unchanged in the digital twin (Fig 5A and 5B). That is, the digital twin predicted that the *Nodal* expression pattern can be maintained without Zfpm, if the *Nodal cis*-regulatory region is properly re-designed. To substantiate this prediction, we tried to manipulate the upstream region of *Nodal*.

Zfpm is a co-factor of Gata.a, but it has not completely been understood how Zfpm-dependency is encoded in *cis*-regulatory regions. Therefore, we first tried to find Gata.a binding sites that act independently of Zfpm. *Zic-r.b* is one of Gata.a targets (see Fig 4). When we hypothetically added regulation by Zfpm to all conjunctive clauses that included Gata.a in the *Zic-r.b* RF, this hypothetical RF failed to induce expression in B6.4 cells (Fig 5B). Because expression of *Zic-r.b* in B6.4 is important for muscle cell specification [36,37], this simulation indicated that the muscle specification program in the B6.4 lineage would be disrupted if mutations placed *Zic-r.b* under control of Zfpm. This observation suggested that *Zic-r.b* is constrained to be independent of Zfpm. Indeed, knockdown of *Zfpm* did not affect *Zic-r.b* expression in real embryos (Fig 5C). Therefore, it is highly likely that Gata.a binding sites responsible for *Zic-r.b* expression in 32-cell embryos function independently of Zfpm. Specifically, because two Gata.a-binding sites responsible for *Zic-r.b* expression in B6.4 have been identified [14], it is highly likely that these two Gata.a binding sites act independently of Zfpm.

The *cis*-regulatory region responsible for expression of *Nodal* in b6.5 of 32-cell embryos has been identified and contains four Gata-binding sites [19,38] (Fig 5D). A reporter construct that contained this region and a minimal promoter was expressed in b6.5 (Fig 5E) [19,38]. Because co-injection of this wild-type reporter and the *Zfpm* MO greatly reduced the percentage of embryos with reporter expression (Fig 5E and 5E'), this reporter reproduced the dependency of *Nodal* on Zfpm.

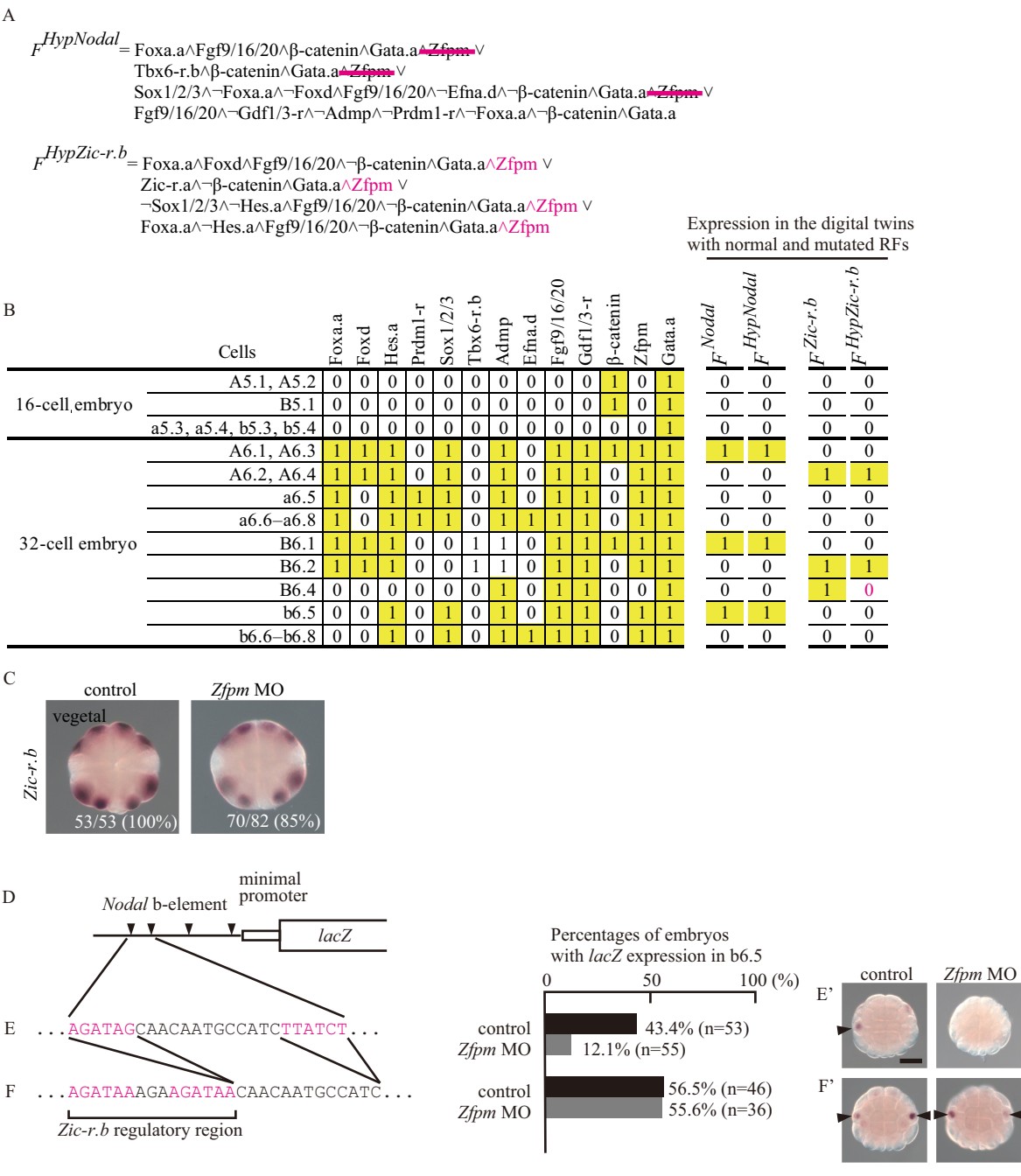

A

$$F^{HypNodal} = \text{Foxa.a} \wedge \text{Fgf9/16/20} \wedge \beta\text{-catenin} \wedge \text{Gata.a} \wedge \cancel{\text{Zfpm}} \vee$$
$$\text{Tbx6-r.b} \wedge \beta\text{-catenin} \wedge \text{Gata.a} \wedge \cancel{\text{Zfpm}} \vee$$
$$\text{Sox1/2/3} \wedge \neg\text{Foxa.a} \wedge \neg\text{Foxd} \wedge \text{Fgf9/16/20} \wedge \neg\text{Efna.d} \wedge \neg\beta\text{-catenin} \wedge \text{Gata.a} \wedge \cancel{\text{Zfpm}} \vee$$
$$\text{Fgf9/16/20} \wedge \neg\text{Gdf1/3-r} \wedge \neg\text{Admp} \wedge \neg\text{Prdm1-r} \wedge \neg\text{Foxa.a} \wedge \neg\beta\text{-catenin} \wedge \text{Gata.a}$$

$$F^{HypZic\text{-}r.b} = \text{Foxa.a} \wedge \text{Foxd} \wedge \text{Fgf9/16/20} \wedge \neg\beta\text{-catenin} \wedge \text{Gata.a} \wedge \text{Zfpm} \vee$$
$$\text{Zic-r.a} \wedge \neg\beta\text{-catenin} \wedge \text{Gata.a} \wedge \text{Zfpm} \vee$$
$$\neg\text{Sox1/2/3} \wedge \neg\text{Hes.a} \wedge \text{Fgf9/16/20} \wedge \neg\beta\text{-catenin} \wedge \text{Gata.a} \wedge \text{Zfpm} \vee$$
$$\text{Foxa.a} \wedge \neg\text{Hes.a} \wedge \text{Fgf9/16/20} \wedge \neg\beta\text{-catenin} \wedge \text{Gata.a} \wedge \text{Zfpm}$$

Expression in the digital twins with normal and mutated RFs

B

| | Cells | Foxa.a | Foxd | Hes.a | Prdm1-r | Sox1/2/3 | Tbx6-r.b | Admp | Efna.d | Fgf9/16/20 | Gdf1/3-r | β-catenin | Zfpm | Gata.a | $F^{Nodal}$ | $F^{HypNodal}$ | $F^{Zic\text{-}r.b}$ | $F^{HypZic\text{-}r.b}$ |
|---|---|---|---|---|---|---|---|---|---|---|---|---|---|---|---|---|---|---|
| 16-cell embryo | A5.1, A5.2 | 0 | 0 | 0 | 0 | 0 | 0 | 0 | 0 | 0 | 0 | 1 | 0 | 1 | 0 | 0 | 0 | 0 |
| | B5.1 | 0 | 0 | 0 | 0 | 0 | 0 | 0 | 0 | 0 | 0 | 1 | 0 | 1 | 0 | 0 | 0 | 0 |
| | a5.3, a5.4, b5.3, b5.4 | 0 | 0 | 0 | 0 | 0 | 0 | 0 | 0 | 0 | 0 | 0 | 0 | 1 | 0 | 0 | 0 | 0 |
| 32-cell embryo | A6.1, A6.3 | 1 | 1 | 1 | 0 | 1 | 0 | 1 | 0 | 1 | 1 | 1 | 1 | 1 | 1 | 1 | 0 | 0 |
| | A6.2, A6.4 | 1 | 1 | 1 | 0 | 1 | 0 | 1 | 0 | 1 | 1 | 0 | 1 | 1 | 0 | 0 | 1 | 1 |
| | a6.5 | 1 | 0 | 1 | 1 | 1 | 0 | 1 | 0 | 1 | 1 | 0 | 1 | 1 | 0 | 0 | 0 | 0 |
| | a6.6–a6.8 | 1 | 0 | 1 | 1 | 1 | 0 | 1 | 1 | 1 | 1 | 0 | 1 | 1 | 0 | 0 | 0 | 0 |
| | B6.1 | 1 | 1 | 1 | 0 | 0 | 1 | 1 | 0 | 1 | 1 | 1 | 1 | 1 | 1 | 1 | 0 | 0 |
| | B6.2 | 1 | 1 | 1 | 0 | 0 | 1 | 1 | 0 | 1 | 1 | 0 | 1 | 1 | 0 | 0 | 1 | 1 |
| | B6.4 | 0 | 0 | 0 | 0 | 0 | 0 | 1 | 0 | 1 | 1 | 0 | 0 | 1 | 0 | 0 | 1 | 0 |
| | b6.5 | 0 | 0 | 1 | 0 | 1 | 0 | 1 | 0 | 1 | 1 | 0 | 1 | 1 | 1 | 1 | 0 | 0 |
| | b6.6–b6.8 | 0 | 0 | 1 | 0 | 1 | 0 | 1 | 1 | 1 | 1 | 0 | 1 | 1 | 0 | 0 | 0 | 0 |

C

control *Zfpm* MO

*Zic-r.b* vegetal

53/53 (100%) 70/82 (85%)

D

*Nodal* b-element minimal promoter lacZ

Percentages of embryos with *lacZ* expression in b6.5

0 50 100 (%)

E ...AGATAGCAACAATGCCATCTTATCT...

control 43.4% (n=53)
*Zfpm* MO 12.1% (n=55)

F ...AGATAAAGAAGATAACAACAATGCCATC...

*Zic-r.b* regulatory region

control 56.5% (n=46)
*Zfpm* MO 55.6% (n=36)

E'
control *Zfpm* MO

F'

**Fig 5. The digital twin illustrates how the regulatory region of *Nodal* can become independent of *Zfpm*.** (A) Regulation by Zfpm is hypothetically removed from the *Nodal* RF (magenta lines) and added to conjunctive clauses with Gata.a in the RF of *Zic-r.b* (magenta letters). (B) The distribution of upstream factors involved in expression of *Nodal* and *Zic-r.b* is shown on the left. On the right, expression of *Nodal* and *Zic-r.b* in 16-cell and 32-cell digital twins with normal ($F^{Nodal}$ and $F^{Zic\text{-}r.b}$) and hypothetically mutated RFs ($F^{HypNodal}$ and $F^{HypZic\text{-}r.b}$) is shown. *Zic-r.b* expression was lost in B6.4 in the digital twin with mutated RFs (magenta), while *Nodal* expression was not changed. (C) Expression of *Zic-r.b* was examined with *in situ* hybridization in normal and *Zfpm* morphant embryos. Total numbers of embryos examined and numbers of embryos that photographs represent are shown within the panels. (D) The upstream region of *Nodal* [nucleotide position 6168259 to 6168410 of Chromosome 14; HT version of the assembly [46]], which promotes expression in b6.5 [38], contains four putative Gata.a sites (triangles). A reporter construct that contains this region is expressed in b6.5 cells of 32-cell embryos [38]. (E, F) Expression of *lacZ* reporter constructs was examined with *in situ* hybridization at the 32-cell stage, and embryos with *lacZ* expression in b6.5 were counted. (E) Co-injection of the wild-type reporter construct with the *Zfpm* MO greatly reduced expression of reporter genes. (F) Co-injection of the *Zfpm* MO did not reduce expression of a mutant construct in which two upstream Gata.a binding sites were removed and the *Zic-r.b* upstream regulatory region containing Gata.a binding sites was inserted. Arrowheads in (E') and (F') indicate expression of *lacZ* in b6.5. Note that not all b6.5 cells express *lacZ* because of mosaic incorporation of *lacZ* constructs. Scale bar, 50μm.

Next, we deleted the two upstream Gata.a binding sites of the *Nodal cis*-regulatory region, and instead inserted the two Gata.a binding sites responsible for *Zic-r.b* expression. This construct was expressed in b6.5, but co-injection of the *Zfpm* MO did not reduce the percentage of embryos with reporter expression (Fig 5F and 5F'). Thus, by this replacement, the reporter became independent of Zfpm, and the mutated upstream sequence retained the ability to direct expression in b6.5. Because *Zic-r.b* is not expressed in b6.5, it is not likely that the Gata.a binding sites of *Zic-r.b* give temporal and spatial information sufficient for driving the reporter in b6.5. That is, we substantiated the prediction that rewiring to make *Nodal* independent of Zfpm does not necessarily alter the *Nodal* expression pattern, and demonstrated that the observed redundancy of Zfpm for *Nodal* regulation is not an artifact.

## Conclusions

We created a digital twin that accurately reproduces expression dynamics of 13 genes that initiate expression in ascidian 32-cell embryos. These RFs are deduced using data from a comprehensive *in situ* hybridization assay for regulatory genes [7] and exhaustive knockdown assays in both a previous study [5] and the present study. Therefore, we expect that we succeeded in mathematically representing the whole regulatory system of regulatory genes that initiate expression at the 32-cell stage, although there is a small possibility that additional hypothetical redundant factors are found in future. On the other hand, many regulatory genes analyzed in the present study are also expressed in later stages of the life cycle [7]. Such expression will be regulated by different mechanisms, and these mechanisms may not be included in the RFs we used in the present study. Nevertheless, we showed that this digital twin is useful for predicting expression patterns in various experimental conditions in early embryos, and indeed created the pattern that is normally seen in a neural cell pair (b6.5) of the 32-cell embryo in the 16-cell embryo. Thus, the RFs predict gene expression patterns under untested conditions; therefore, the RFs are not mere restatements of experimental results.

We showed that simulators that digitally reproduce gene expression patterns are useful for analyzing mechanisms that are largely inaccessible by experiments using real embryos. Our simulator revealed that regulation that is theoretically redundant or dispensable to give specific expression information occurred in more than half the cases involving the 13 RFs. This indicates robust buffers in the gene regulatory network. Robust buffers may have created a basis for GRN rewiring that changes expression patterns. Although it is unknown whether only the *Ciona* GRN in early embryos has such wide tolerance or whether this is a general feature of GRNs, we propose that such rewiring of GRNs without changing expression patterns, i.e. developmental system drift, may have occurred frequently during evolution. Indeed, examples of developmental system drift have been reported in ascidians [31,39,40]. If so, it is possible that GRNs in ancestral animals may have had more redundancy and that such redundancy may have decreased during evolution.

## Materials and methods

### Animals and gene identifiers

Adult specimens of *Ciona intestinalis* (type A; also called *Ciona robusta*) were obtained from the National BioResource Project for *Ciona*. cDNA clones were obtained from our EST clone collection [41]. Identifiers for genes examined in this study are as follows: *Admp*, KY21. Chr2.381/KH.C2.421; *Bmp3*, KY21.Chr12.897/KH.C12.491; *β-catenin* (*Ctnnb*), KY21.Chr9.48/ KH.C9.53; *Dickkopf*, KY21.Chr6.647/KH.L20.29; *Dlx.b*, KY21.Chr7.361/KH.C7.243; *Dmrt.a*, KY21.Chr5.707/KH.S544.3; *Efna.d*, KY21.Chr3.881/KH.C3.716; *Ets1/2* (*Ets1/2.b*), KY21. Chr10.346/KH.C10.113; *Fgf9/16/20*, KY21.Chr2.824/KH.C2.125; *Foxa.a*, KY21.Chr11.1129/

KH.C11.313; *Foxd*, KY21.Chr8.654/655/KH.C8.396/890; *Foxtun2*, KY21.Chr14.884/KH.L150.2; *Gata.a*, KY21.Chr6.630/KH.L20.1; *Gdf1/3-r*, KY21.Chr4.347/KH.C4.547; *Hes.a*, KY21.Chr1.29/KH.C1.159; *Hes.b*, KY21.Chr3.564/KH.C3.312; *Lhx3/4*, KY21.Chr13.457/KH.S215.4; *Macho-1 (Zic-r.a)*, KY21.Chr1.1337/KH.C1.727; *Neurogenin*, KY21.Chr6.434/KH.C6.129; *Nodal*, KY21.Chr14.864/KH.L106.16; *Otx*, KY21.Chr4.720/KH.C4.84; *Pem1*, KY21.Chr1.618/KH.C1.755; *Prdm1-r*, KY21.Chr12.994/997/KH.C12.105/493; *Raf*, KY21.Chr1.1417/KH.L18.20; *Snail (Snai)*, KY21.Chr3.1356/KH.C3.751; *Sox1/2/3*, KY21.Chr1.254/KH.C1.99; *Tbx6-r.a*, KY21.Chr11.458/KH.L8.11; *Tbx6-r.b*, KY21.Chr11.465/466/467/KH.S654.1/2/3; *Tcf7*, KY21.Chr6.59/KH.C6.71; *Tfap2-r.b*, KY21.Chr7.1145/KH.C7.43; *Wnt3*, KY21.Chr9.971/KH.C9.27; *Wnt5*, KY21.Chr4.1174/KH.L152.45; *Wnttun5*, KY21.Chr9.822/KH.C9.257; *Zfpm (Fog)*, KY21.Chr10.450/KH.C10.574; *Zic-r.b*, KY21.Chr6.26/27/28/29/30/31/KH.S816.1/2/4/KH.L59.1/12. Identifiers for the latest KY21 set [42] and a more commonly used KH set [43] are shown.

## Functional assays

The sequence of the MO for *Zfpm*, which blocks translation, was 5'- GGACATTGTGTGTGT TATTTTTGTA -3'. *Gata.a*, *β-catenin*, and *Efna.d* MOs that were used in previous studies [14,17,21,26] were used here. These MOs were microinjected under a microscope. For overexpression, coding sequences of *Zfpm*, *Sox1/2/3*, *Tbx6-r.b*, *Foxa.a*, and *Prdm1-r* were cloned into pBluescript RN3 [44]. While there are two copies of *Prdm1-r*, we used *Prdm1-r.a* (KY21.Chr12.997) for overexpression. Injected RNAs were transcribed using a mMESSAGE mMACHINE T3 Transcription Kit (Thermo Fisher Scientific, #AM1348). To stabilize β-catenin in all cells, embryos were incubated in sea water containing BIO, which is an inhibitor of Gsk3 (Merck, #361550; 2.5 μM) [13], from the late 16-cell stage.

To mutate *Zfpm*, we used CRISPR technology. A guide RNA (5'-CCACGTCTCACCTGAA AGATATC-3') was synthesized *in vitro* using a precision gRNA synthesis kit (Thermo Fisher Scientific, #A29377), and the guide RNA (200 ng/μL) and Cas9 protein (50 ng/μL; Thermo Fisher Scientific, #A36496) were co-injected. For genotyping, injected embryos were lysed and genomic DNA was extracted. Then, we amplified a genomic region that contained the target site by PCR (primer sequences are: 5'-CCATTCCGAACTTTCGTCGC-3' and 5'-CGCT GCTTTTGTCATGTGGT-3'), and amplified DNA fragments were subjected to sequencing analysis. CRISPR efficiency was estimated with obtained sequence chromatograms and the TIDE program [45].

To mimic overexpression of *Fgf9/16/20*, we added human recombinant basic FGF (FGF2; Merck, #662005) to sea water with 0.1% BSA at a concentration of 10 ng/mL. All functional assays were performed at least twice with different batches of embryos.

## Reporter assays

We used a reporter construct from a previous study [19] as a starting construct. This construct contained an upstream region of *Nodal* [genomic coordinates are Chr14:6168259–6168410 in the HT version assembly [46]], a basal promoter region of *Zfpm*, and *lacZ*; therefore, the overall configuration is the same as that of the construct used in another study [38]. Reporter constructs were introduced into unfertilized eggs by microinjection. Expression of *lacZ* was examined by *in situ* hybridization. Reporter experiments were performed at least twice with different batches of embryos.

## Gene expression assays

Whole-mount *in situ* hybridization was performed, as described previously [5]. For fluorescent detection, we used Tyramide SuperBoost Kits (Thermo Fisher, #B40922). For RT-qPCR, we

extracted RNA and converted RNA to cDNA using a Cells-to-Ct kit (Thermo Fisher Scientific, #4402954). cDNA samples were analyzed by quantitative PCR with the SYBR-Green method. Used primers were: *Nodal*, 5′-GGAATTGTACCGAGCCAAAA-3′ and 5′- ACGACGACCAACTTTGAACC-3′; *Lhx3/4*, 5′-GGTTGGCAAATGGAAGTCGAA-3′ and 5′-GCCAAGGTTTGTCCTGTACTTTGAG-3′; *Neurogenin*, 5′-GGCCTCACAAGACGTAATGG-3′ and 5′-AAGACCATGCATTCGGTTTC-3′; *Dickkopf*, 5′-ACACCTACTATAATACCTAAACGCGAAA-3′ and 5′-TTGTGCGCAACAGAAACCAT-3′; *Snail*, 5′-TGGTAAAGCGTTCTCACGTACCT-3′ and 5′-CACAGTGCATTGGTATGGTTTCTC-3′; *Wnt3*, 5′-ATTGACCAATGCAAGCATCA-3′ and 5′-TCCAATACAGGCCCGAATAC-3′; *Wnt5*, 5′-ATCGGGAACGTAAAGTAATGAACAT-3′ and 5′-CGAGCCGATCTCACAACGA-3′; *Bmp3*, 5′-GTCCGTAGCTTCTTCTCTGTAGCA-3′ and 5′-GCGGGTACGATTAGAATAGGTTTC-3′; *Hes.b*, 5′-CTTCGACTGTGCAAATTGTATCTTC-3′ and 5′-CGCGGCGTCGTTTTTC-3′; *Zic-r.b*, 5′-CGTTTGGAAGAAGCGAGAATTTAA-3′ and 5′-TTCAGTGTTGTGCATGTAACTATGCTT-3′; *Dmrt.a*, 5′-TCTGATCGCTGAACGACAAC-3′ and 5′-GTGGCGACTGTCGGTTATTT-3′; *Otx*, 5′-GGCTTAGGCCACGATATGAA-3′ and 5′-TAGCTCCTTGGTGCATTCCT-3′; *Dlx.b*, 5′-TTACAAACTGCACCCCCTTC-3′ and 5′-TCTCCTGGATCGGAATCAAC-3′; *Macho-1*, 5′- CCCAGTATGCACCAAATTCAGA-3′ and 5′- TGGTGTGAAAACGGGTGAAAC-3′; *Pou2*, 5′-AAGATGGTTGCTGGATGCTAATAAT-3′ and 5′-TTGGATTGGAGTGGGAATAACAA-3′. No amplification was observed in control samples that included water instead of reverse-transcriptase.

## Boolean representation of *Nodal* regulatory function and the digital twin

To identify the simplest disjunctive normal form compatible with all experimental results in the present study and our previous study [5], we used a computer program, mindnf, that we developed previously [5].

The digital twin, in which RFs for the 13 genes are implemented, is provided as an HTML-based program (http://ghost.zool.kyoto-u.ac.jp/sim32v2/). The computer code is deposited in Zenodo (doi: 10.5281/zenodo.7604201).

## Supporting information

**S1 Fig. The expression pattern of *Zfpm*.** Fluorescence *in situ* hybridization was used to examine expression patterns of *Zfpm* in (A) 8-cell, (B) 16-cell, and (C) 32-cell embryos. Note that expression is detected as green dots in nuclei of cells in the animal and vegetal hemispheres of a 16-cell embryo and in the animal hemisphere of a 32-cell embryo, but not in the 8-cell embryo or in the vegetal hemisphere of the 32-cell embryo. Nuclei are stained with DAPI (blue).
(PDF)

**S2 Fig. CRISPR-based knockout of *Zfpm* to confirm the specificity of the phenotype in *Zfpm* morphants.** (A) First, we examined genomic DNA of larvae developed from eggs injected with Cas9 protein and a guide RNA designed to bind to the region encoding the fifth and sixth zinc fingers of Zfpm. Sequencing followed by TIDE analysis [45] indicated that 69.6 and 74.4% of DNA fragments amplified with PCR from two batches of larvae contained mutations, suggesting that this guide RNA was effective. Next, we performed the same experiments using 16-cell embryos. Mutagenic efficiency varied from 1.1 to 69.5% among nine batches of embryos we examined. Therefore, we used three batches of embryos that showed high mutagenic efficiencies (69.5, 52.9, and 45.6%) for the following analyses. TIDE further indicated that these three batches contained 3/6/9 base insertions or deletions in 11.2%, 12.9%, and 6.1% of the amplified DNA fragments. Because these mutations did not cause frame-shifts, and may

not have severely impaired *Zfpm* function, we conservatively estimated that 58.3 (= 69.5–11.2), 40.0 (= 52.9–12.9), and 39.5 (= 45.6–6.1) % of these embryos contained effective mutations. As all cells are diploid, 34.0, 16.0, and 15.6% of cells were estimated to contain effective mutations in both maternal and paternal alleles. By *in situ* hybridization, we found that 20.7, 18.9, and 10.0% of embryos in these batches lost *Nodal* expression. These percentages were close to the expected percentages, indicating that *Zfpm* is required for *Nodal* expression and that the *Zfpm* MO acted specifically. (B) An embryo that lost *Nodal* expression by CRISPR knockout of *Zfpm*. Expression was examined with *in situ* hybridization. An uninjected embryo (n = 43) and embryos injected with either Cas9 (n = 45) or *Zfpm* sgRNA (n = 39) are shown as controls. *Nodal* expression was not changed in these controls.
(PDF)

**S3 Fig. The tentative *Nodal* regulatory function.** This tentative function was determined previously [5], and explains the *Nodal* expression pattern in normal conditions at the 16-cell and 32-cell stages and in a variety of experimental conditions at the 32-cell stage, but cannot accurately predict expression patterns in experimental conditions at the 16-cell stage. Note that expression patterns of Prdm1-r and Foxa.a indicated that either of them or both are involved in the fourth conjunctive clause, but their involvement was not strictly tested [5].
(PDF)

**S4 Fig. Examination of the fourth conjunctive clause of the RF for *Nodal*.** *Nodal* expression was examined with *in situ* hybridization at the 16-cell stage in (A) an embryo treated with FGF2, (B) an embryo injected with the *Zfpm* MO and treated with FGF2, (C) an embryo injected with the *Gata.a* MO and treated with FGF2, (D) an embryo injected with the *β-catenin* MO and treated with FGF2, (E) an embryo injected with *Prdm1-r* mRNA and treated with FGF2, (F) an embryo injected with *Foxa.a* mRNA and treated with FGF2. *Nodal* expression in unperturbed control embryos is shown in Fig 3D. Photographs in (A) are the same as those in Fig 2C. Total numbers of embryos examined and numbers of embryos that photographs represent are shown within the panels.
(PDF)

**S5 Fig. Rewiring that does not change expression patterns in normal embryos can induce different expression patterns if distribution patterns of upstream regulators are changed.** (A) RFs for *Lhx3/4*, *Neurogenin*, and *Dickkopf* are commonly represented as Foxd∧Fgf9/16/20∧β-catenin, and these genes are expressed in A6.1, A6.3, and B6.1 of normal embryos. Regulatory functions represented as Fgf9/16/20∧β-catenin and Foxd∧β-catenin can induce the same expression patterns. (B-D) In cases in which (B) Foxd, (C) Fgf9/16/20, or (D) β-catenin acts in all cells of 16-and 32-cell embryos, three RFs induce different expression patterns. This observation indicates that *Lhx3/4*, *Neurogenin*, and *Dickkopf* could be expressed differently upon changes of distribution patterns of their upstream factor through rewiring that does not change expression patterns in normal embryos.
(PDF)

**S6 Fig. β-catenin and Gata.a are involved in regulating *Dmrt.a* and *Dlx.b* expression.** (A) *Dmrt.a* expression in normal unperturbed embryos. (B) *Dmrt.a* expression is abolished in embryos incubated in sea water containing BIO, an inhibitor of Gsk3; therefore, β-catenin is expected to be stabilized in all cells. (C) *Dmrt.a* expression is also abolished in embryos injected with the *Gata.a* MO. (D) *Dlx.b* expression is detected in anterior cells of the animal hemisphere of embryos injected with a MO against *Efna.d*, as we reported before [5]. (E) *Dlx.b* expression is not detected in embryos injected with the *Efna.d* MO and incubated in sea water containing BIO. (F) *Dlx.b* expression was not detected in embryos injected with the *Efna.d*

MO and the *Gata.a* MO. Arrowheads indicate maternal *Dlx.b* mRNA localized to the posterior pole. Total numbers of embryos examined and numbers of embryos that photographs represent are shown below.
(PDF)

**S1 Table. All experimental conditions and experimental results used to determine the *Nodal* regulatory function.**
(XLSX)

**S2 Table. Regulatory functions for genes that initiate expression at the 32-cell stage.**
(XLSX)

## Acknowledgments

We thank Reiji Masuda, Shinichi Tokuhiro, Chikako Imaizumi (Kyoto University), Manabu Yoshida (University of Tokyo), and other members working under the National BioResource Project for *Ciona* (MEXT, Japan) at Kyoto University and the University of Tokyo for providing experimental animals. We also thank Steven D. Aird for editing the manuscript.

## Author Contributions

**Conceptualization:** Yutaka Satou.

**Formal analysis:** Miki Tokuoka.

**Investigation:** Miki Tokuoka.

**Writing – original draft:** Yutaka Satou.

**Writing – review & editing:** Miki Tokuoka, Yutaka Satou.

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
