## [Decision Letter · Decision Letter 0]

6 Jun 2023

Dear Dr Satou,

Thank you very much for submitting your Research Article entitled 'A digital twin reproducing the gene regulatory network of early Ciona embryos indicates robust buffering in the network' to PLOS Genetics.

The manuscript was fully evaluated at the editorial level and by three independent peer reviewers. We apologize for the length of this process. The reviewers appreciated your work. Referee 2 supports its publication, but referees 1 and 3 raised substantial concerns about the current manuscript. Referee 3, in particular, is worried that the way you constructed the networks in this and in your previous study (Tokuoka et al., 2021) may artificially favour the selection of redundant networks, which would undermine a major message of your study. Referee 1, besides encouraging you to revise/precise many sentences, is worried that the use of boolean networks may provide little more insight than providing a mere restatement of experimental results. 

Based on these significant concerns, we will not be able to accept this version of the manuscript, but we would be willing to review a much-revised version that in particular addresses these two major concerns. We cannot, of course, promise publication at that time.

If you decide to revise the manuscript for further consideration at PLOS Genetics, please aim to resubmit within the next 60 days, unless it will take extra time to address the concerns of the reviewers, in which case we would appreciate an expected resubmission date by email to plosgenetics@plos.org.

To enhance the reproducibility of your results, we recommend that you deposit your laboratory protocols in protocols.io, where a protocol can be assigned its own identifier (DOI) such that it can be cited independently in the future.

We are sorry that we cannot be more positive about your manuscript at this stage. Please do not hesitate to contact us if you have any concerns or questions.

Yours sincerely,

Patrick Lemaire

Guest Editor

PLOS Genetics

Gregory Barsh

Editor-in-Chief

PLOS Genetics

Reviewer's Responses to Questions

**Comments to the Authors:**

Reviewer #1: I have now carefully read this manuscript. My major objection is that the English is, at times, rather hard to read. The article provides an improvement on a previous related article. The improvement consist in the inclusion of one gene, Nodal, and an exploration of the redundancy in the regulation of the patterns of gene expression in the 32 cell stage. There are really not major take-home message rather than a slightly more detailed description of the underlying network (e.g. no actual cell signaling occurring). I am a theoretical biologist and I do not see much benefit in the boolean model. In practice it is just a re-statement of experimental findings. This is certainly not useless and the virtual twin can be helpful as a kind of summary for Ciona researchers. I am uncertain as to whether this manuscript provides enough incremental insight as to deserve publication in PLOS Genetics, I leave that decision to reviewers that work experimentally with Ciona.

I have a number of minor comments. In truth, these are only the major comments among the minor ones because there are quite many grammar mistakes and obscure sentences over the article. The article needs to revised by a native English speaker.

Minor:

-”Although the structure of such networks, which represent connections among

regulatory genes, has been studied extensively, it is unclear how these networks encode gene

expression dynamics.”. It is unclear what “encode” is supposed to mean in this sentence, I think encode is not the right word in this context.

-In line 48, It is just incorrect to label a phylogenetic group as “basal”. Please correct and check: https://pubmed.ncbi.nlm.nih.gov/16733736/

-I think the reading of the article would be facilitated by a better explanation or more detailed discussion of this statement: “Regulatory

mechanisms for 12 of these 13 genes have been described mathematically as Boolean functions,

56which represent necessary and sufficient conditions for their expression (Tokuoka et al., 2021)

(see an example for Boolean functions in Figure 2A).”

-This sentence is not clear enough: “The regulatory function (RF) for the remaining gene, Nodal, was also tentatively determined; it represents necessary conditions, but

does not represent sufficient conditions (Tokuoka et al., 2021). “. What is not necessary or sufficient? And for what?

-This sentence is also unclear: “In the present

study, we first devised such a digital twin and then showed that gene expression patterns can

indeed be manipulated according to predictions from the digital twin.” Are the authors meaning that they are going to perform experimental manipulations? This is clearer later but it should be clear already at this point.

-The following sentence is also unclear: “We

found robust buffering of regulatory mechanisms, which potentially permits rewiring of GRNs

without changing expression patterns.” What is a regulatory mechanism for the authors? How it relates to Rfs. What do they mean by “buffering”, this word is written as a verb form. This suggests that buffering is an active thing in this case but this sound a bit confusing and it is anyway unclear what, if anything, is doing the buffering.

-It is not totally clear what the authors mean by a gene’s RF. The authors provide no explicit definition.

-I cannot understand this statement: “The finding that RFs are

successfully represented as Boolean functions indicates that qualitative, but not quantitative,

control are important for regulation in early ascidian embryos (Tokuoka et al., 2021).” The last part after “control” is objectively not-understandable. This is probably because the authors use control and regulation in non-standard ways, that need to be clarified.

-The grammar in this statement is improvable: “We assumed that cells descended from cells expressing a transcription factor gene at the 16-cell stage express the encoded protein at the 32-cell stage

because of a delay between gene expression and protein translation (Tokuoka et al., 2021).”

-Is “artifactitiously” an actual word? How about just “artificially”

-The one-paragraph sentence between lines 154 and 164 needs to be broken into several different sentences, otherwise it is completely unredeable.

-The title of this section “Reproduction of the pattern of a cell of a 32-cell-embryo in 16-cell embryos” is written in a rather strange manner.

-In line 231: “Because RFs are encoded in cis-regulatory regions,…” Is that the case? Rfs can also be regulated by protein-protein interactions occurring between transcriptional factors and co-factors, as the authors results seem to indicate for the regulation of Nodal.

-”For RFs for the 13 genes, 17 regulators appear 99 times in total.” This sentence is not understandable.

-In line 244: “This finding suggests a possibility that the observed redundancy provides

245a basis for developmental system drifts.” This is statement is true but the reverse can also be true: this redundancy may be the result of developmental systems drift, not just a possible cause.

-In line 246: ”The digital twin also indicated that such rewiring enabled

these genes to behave differently from one another if expression patterns of upstream factors

were changed (S5B-D Fig)” The verb tenses seem to be off in this sentence. From the writing is seems that that is what happened in nature in the past, while that happened only in the model.

-In line 248: ”Although combinations of regulators may not be the only

evolutionary constraints of Rfs,” It is unclear what the authors mean by “combinations of regulators” and “evolutionary constraints” in the context of this sentence.

-I wonder to which extend the robustness results are a mere consequence of not getting the RF right for all genes. Ideally, the reader would not have to read all previous articles by the authors to figure out whether this is the case or not.

Reviewer #2: In this manuscript, Tokuoka and Satou present an updated version of their 16-32-cell stage digital twin embryo. Previously, they unraveled the simplest combinations of “regulatory functions” (RF) for 12 out of 13 regulatory genes/zones, activated at the 32-cell stage, using Boolean logic (Tokuoka et al, 2021). Here, they extend this analysis to cover the 13th gene, Nodal, completing its RF, with the addition of the regulator Zfpm (fog). The authors convincingly tested their predicted Nodal RF (combination of regulatory factors/conditions sufficient for Nodal expression) using the digital twin embryo and real embryo experiments. Next, the authors modified the theoretical RFs for all 13 genes and found that in 55 cases in which a single regulator was removed, the digital twin still predicted the correct gene expression output. In other words, in terms of computation, many factors had ‘redundant’ coding functions. The authors propose the hypothesis that this type of redundant coding function could be a mechanism that would allow developmental systems drift, since changes in regulatory sequences, when coupled with cis-regulatory changes, should still allow the same phenotypic output. Furthermore, the authors then test in embryos the hypothesis that removal of redundant factors will lead to the same phenotypic output if the cis-regulatory sequences are also adapted. They did this by swapping the Gata.a binding sites in the Nodal b6.5 enhancer (dependence on Zfpm) with those of Zic-r.b (independence on Zfpm), which confirmed their hypothesis. The updated HTML-based program is improved and easy to use and will be very useful for the community.

The manuscript is carefully conducted and interpretation is clever and interesting. I think that this thought-provoking manuscript will be of great interest for both ascidian developmental biologists and the broad audience of evo-devo researchers.

Minor comments.

Line 23: Sentence is not clear, I would replace ‘or developmental system drift’ with ‘enabling (or allowing) developmental systems drift’.

line 136: I would remove the word “artifactitiously” (or change it).

Reviewer #3: This is an interesting paper where the authors present a boolean model that is able to recapitulate the WT expression of dynamics of 13 genes in the 32-cell ascidian embryo as well as predict expression in certain experimental conditions. They then do in-silico knockout experiments to conclude that their network is highly redundant, in order to then speculate that it is this redundancy in the network that underlies the evolution of early ascidian patterning via developmental systems drift.

Some clarifications are required throughout, but importantly, in order to say anything about redundancy in the network the authors should first show that the way in which they construct their networks to begin with doesn't inadvertently favour redundant networks. This is particularly important because these networks are constructed taking a bottom-up approach rather than an unbiased parameter fitting approach.

Abstract

L13 (also L27 and L45): How GRNs encode dynamics and evolve are one of the most-studied and best-understood problems in biology, although I do agree that there is still a lot that we don’t know. I would encourage the authors to rephrase.

L14 (and throughout the paper): I find the terminology confusing. Why digital twin? Why not Boolean model of gene regulation?

L21: Playing Devil’s advocate, it might also reflect that the GRN was initially formulated to include many factors that aren’t necessary, and that might not actually be there in the embryo. How do we tease the two possibilities apart?

Introduction

L54-L59: More information on how RF were constructed in Tokuoka 2021 might be beneficial for the reader, particularly since the authors mention explicitly that for Nodal, this represented necessary but not sufficient conditions. The reader needs to know how these functions have been constructed, because the paper’s main claim is one of robustness of the network to evolutionary changes in the regulation. However, is these networks (RFs) are initially built assuming necessity, but with no, or little consideration, for the sufficiency, then it surely comes as no surprise that some of the components were redundant in the formulation to begin with

L80: I think it is important to mention how these RFs were found in thee 2021 paper, in addition to how they are formulated. Why did the algorithms used in Tokuoka 2021 not reveal more minimal RFs, if the simplest RFs were selected?

L97 (also L55): Please clarify why the RF for Nodal is the only one out of 13 that wasn’t calculated or inferred in the previous study (or did I understand this wrong?)

Results

L102: Reading this sentence has made me realise that I don’t quite understand how the simulations are being run. What is being simulated and how are these models initialised. From the introduction, I thought that the models are initialised using the expression at the 16-cell stage to simulate the expression at the 32-cell stage, but the sentence in L102 makes me think I have misunderstood this. Please clarify somewhere in the introduction.

L104: I would have thought that if the model recapitulates the pattern at the 32-cell stage but not at the 16-cell stage, the conclusion would be that the formulation was missing something present at the 16-cell stage, not at the 32-cell stage as stated.

L105: I thin that when the authors say inferred here, they really mean assumed.

L133: The fourth conjunction represents experimental conditions. Which specifically?

L134-L138: I’m not sure what the perturbation is here. Is it FGF up-regulation?

L153: Is the 4th conjunction an aggregate function representing every experimental perturbation then? Please clarify the difference between the 1st to 3rd conjunctions and the 4th somewhere in the introduction, and how these are inferred.

L166-L180: Verry useful recap, thank you!

L180: Except for Foxa.a injections + FGF.

L186: for wild type expression?

L187-L190: I am very confused by these statements. I am also not sure that adding a conjunction to capture experimental perturbations separately from the other 3 is rigorous. Surely in the embryo, the result of the perturbations is a consequence of the normal WT wiring of the GRN; it doesn’t have a separate “perturbation/experimental” module. If the authors want to include this for the sake of the model, OK, but to then from then assume that this conjunction actually exists separately from the main, or WT GRN, and that it can evolve separately and/or be useful later in development, is a stretch.

L196: his statement is incredibly confusing. Is it the case, that the all the simulating is being done at the 16-cell stage, and that they are simulating the presence of mRNAs of different genes in different cells according to whether the proteins of these are expressed in the daughter cells at the 32-cell stage?

L199-L202 clarify the previous point. Please, try to clarify this throughout.

L218: I wouldn’t call this a model prediction. The model has been constructed and selected to reproduce these patterns. Therefore, there is agreement between model and data, but this is not a model prediction.

L222: transformed cells?

L230: Although I agree with the results presented in the section beginning in L230, I think a more detailed discussion regarding how the original construction/inference of the networks might have favoured redundancy in the first place, is required. As I mentioned elsewhere, if the way in which these networks were constructed focused on necessity rather than sufficiency, it is possible that the algorithm had favoured networks with a high degree of redundancy to begin with. This does not mean that the GRNs in the Ascidian early embryo are redundant, in fact, they might not be, as the minimal GRN in this section shows. I don’t think this is the case, and think that the authors are probably right, but a comment on whether their initial model construction favoured, or at least did not discard, redundancy the networks is needed in order for the results presented in this section to be more rigorous.

L233: perturb?

L260: But you have shown above that Zfpm positively regulates Nodal (L126). Please clarify.

L256: I wonder whether the the flow of the paper might be improved if this section was to be moved to follow the section on the regulatory function of nodal?

Conclusions

L298: I think that here the authors are over-claiming slightly: Having all the data does not reveal the underlying the regulatory structure, hence the motivation for the modelling. I would rephrase to reflect this.

L306: The conclusion needs an in-depth exploration of whether the inferred networks might have had landed on more redundant formulations by default, or by design of the algorithms used to infer them. This is my biggest criticism of this paper: that the main claim – namely that redundancy within the GRN might have led to developmental systems drift in ascidians – is due to the networks in the early embryos of one extant species of ascidian being very redundant, where these networks have been inferred by constructing Boolean models that reproduce the data from the bottom up (not using unbiased parameter fitting or reverse-engineering for example).

**Have all data underlying the figures and results presented in the manuscript been provided?**

Reviewer #1: Yes

Reviewer #2: Yes

Reviewer #3: Yes

PLOS authors have the option to publish the peer review history of their article (what does this mean?). If published, this will include your full peer review and any attached files.

Reviewer #1: No

Reviewer #2: No

Reviewer #3: No

---

## [Decision Letter · Decision Letter 1]

16 Aug 2023

Dear Dr Satou,

Thank you very much for submitting your Research Article entitled 'A simulator reproducing the gene regulatory network of early *Ciona* embryos indicates robust buffers in the network' to PLOS Genetics.

The manuscript was fully evaluated at the editorial level and by independent peer reviewers. Following your revisions, Reviewer 3 joined Reviewer 1 in supporting publication. As Reviewer 2 was not available to assess the revisions, we asked Reviewer 4, who was not involved in the first round of refereeing, to comment on your work.

Reviewer 4 appreciated the attention to an important topic and identified some concerns with the model that we ask you address in a revised manuscript. Upon resubmission, the manuscript will not be sent back out for review.

We suggest that you give particular attention to the following two comments:

1: “In any case, for Lhx3/4, Neurogenin, and Dickkopf, it would be interesting to check if the function (Foxd & β-catenin)v(Fgf9/16/20 & β-catenin) was discarded because of the choice decided by the authors, or because it didn't comply with some experiments.”

2: “ how can we ensure, if a DNF with n literals is choosen, that we are not missing an additional regulator? e.g. if (AvB) as well as (AvBv¬C) were compatible with the experiments disclosing expression of X, it could be the case that in the future, a new experiment shows that indeed absence of C is required for the expression of X.”

Additionally, we encourage you to replace "conjunctions" with "conjunctive clauses" when applicable, to provide the model (the set of functions) in the form of an excel sheet or, better, an SBML file and to revise the title of your study, possibly following Reviewer 3's suggestion to include "digital twin".

Finally, we ask that you:

Yours sincerely,

Patrick Lemaire

Guest Editor

PLOS Genetics

Gregory Barsh

Editor-in-Chief

PLOS Genetics

Reviewer's Responses to Questions

**Comments to the Authors:**

Reviewer #3: I thank the authors for addressing my main concern, which was that their networks might be biased towards redundancy by construction and am happy with their response and amendments to the text. I also thank their responses to my other comments. Since writing my review, it has come to my attention that the notion of a “digital twin” is gaining traction. The authors should use it instead of simulator if they wish to. I am now supportive of the publication of this paper.

As an aside, I emphatically agree with the authors that the fact that the model is Boolean, does not make it “a mere re-statement of experimental results”. Boolean models have a lot of explanatory and predictive power, often more so than continuous models, and are perfectly suitable for this system where expression is on/off as opposed to graded.

Ps. The amendment in L356-357 should read “in ancestral animals”.

Reviewer #4: This manuscript presents a program enabling the determination of expression patterns of 13 genes in 32-cell ascidian embryos from that of 18 upstream factors, which start to be expressed in the 16-cell embryo. Regulatory mechanisms are represented as Boolean functions determined in a previous work [Tokuoka et al., 2021] for 12 of these genes. The case of Nodal is revisited as the function determined in [Tokuoka et al., 2021] could not account for Nodal expression in specific situations.

Contributions of this work are the implementation of the computational tool (called simulator), a regulatory function of Nodal, and an in silico experiment to supposedly show that some regulators are redundant. Overall, the paper is interesting, but in my opinion the model definition is not convincing.

- Concerning the implemented program, it is indeed well designed with its HTML version, and might be useful for the community. However, it is a bit excessive to call it a "simulator", as it merely performs the evaluation of Boolean functions (i.e. one step, from an input pattern to a resulting pattern).

- I am not qualified to evaluate the results concerrning the regulatory function of Nodal.

- My main concern relates to the RFs definition and the take home message about a supposed redundancy of many regulatory genes. The authors recognize that while "although more than half the regulators give theoretically redundant temporal or spatial information to target genes, they are necessary for expression of their target genes in real embryos." This statement sounds a bit odd to me, as I would understand "redundancy" differently.

But I am more concerned about the following claim "It is unlikely that the observed redundancy is an artifact derived from our method to determine RFs."

The problem to determine the regulatory functions is largely underdetermined as there are 2^18 (=262,144) potential truth tables for each target gene. As rightly put by the authors, it is impossible to perform so many experiments to determine the right function. When they say "we considered all theoretically possible conjunctions to determine RFs", it might be the case (note that the term "conjunctions" here is not appropriate), but they finally choose a single one, and the redundancy analysis on this sole function is therefore not convincing.

First, criteria for this choice are unclear: "DNFs with the smallest number of conjunctions and the smallest number of upstream regulators were considered as primary candidates." I suppose that the authors by "conjunctions" mean "conjunctive clauses", which should be corrected. Otherwise, it would be unclear which function would be choosen between A&B&C (A and B and C) and (A&B) v (B&C) ((A and B)or(B and C), as they have the same number of literals (A,B,C) and the same number of conjunctions. The notion of simplest function here is debatable, one could say that the simplest function is the least stringent (higher number of disjunctions, i.e. clauses), as AvB is true for 3 value configurations of A and B, whereas A&B is true for only one.

In any case, for Lhx3/4, Neurogenin, and Dickkopf, it would be interesting to check if the function (Foxd & β-catenin)v(Fgf9/16/20 & β-catenin) was discarded because of the choice decided by the authors, or beacuse it didn't comply with some experiments.

Second, how can we ensure, if a DNF with n literals is choosen, that we are not missing an additional regulator? e.g. if (AvB) as well as (AvBv¬C) were compatible with the experiments disclosing expression of X, it could be the case that in the future, a new experiment shows that indeed absence of C is required for the expression of X.

Further comments:

In Nodal's function, I am surprised to see that β-catenin appears as an activator and an inhibitor. Such cases are not so common. It would be good to check again an explain why β-catenin acts as a dual regulator.

I suppose that there exist data on binding sites in ascidian, and it would be interesting to use these data to help choosing the RFs (choice which is so far based on expression data).

In Table S1, it appears clearly that Foxd, Fgf9/16/20 and β-catenin do not have the same patterns of expression in the 16-cell and the 32-cell embryo. This would imply that they are regulated, but I am not sure the authors provide any comment on this point, neither on potential cell-cell signaling. I also wonder if there are some cross regulations, or circuits, as the proposed network is free of loops.

Finally, Boolean functions to model regulatory networks driving development have been used for several decades now. There are many studies, for example L. Sánchez & D. Thieffry as well as R. Albert & HG Othmer modelling work on the regulatory modules controlling drosophila segmentation, ER Álvarez-Buylla papers on Arabidopsis development.

There are also software tools devoted to Boolean models (GINsim, BoolNet,...).

I would advise to provide the model (the set of functions) in the form of an excel sheet or, better, as an SBML file.

Overall, the manuscript would deserve a carefull re-reading as there are some ill-constructed sentences.

The title should be changed. A simulator does not reproduce a network, rather its dynamics. Moreover, as previously mentionned the term "simulator" does not seem appropriate here.

**Have all data underlying the figures and results presented in the manuscript been provided?**

Reviewer #3: Yes

Reviewer #4: Yes

PLOS authors have the option to publish the peer review history of their article (what does this mean?). If published, this will include your full peer review and any attached files.

Reviewer #3: No

Reviewer #4: No

---

## [Editor Report · Decision Letter 2]

1 Sep 2023

Dear Dr Satou,

We are pleased to inform you that your manuscript entitled "A digital twin reproducing gene regulatory network dynamics of early Ciona embryos indicates robust buffers in the network" has been editorially accepted for publication in PLOS Genetics. Congratulations!

Yours sincerely,

Patrick Lemaire

Guest Editor

PLOS Genetics

Gregory Barsh

Editor-in-Chief

PLOS Genetics

Comments from the reviewers (if applicable):

**Data Deposition**

http://datadryad.org/submit?journalID=pgenetics&manu=PGENETICS-D-23-00299R2

**Press Queries**

---

## [Editor Report · Acceptance letter]

15 Sep 2023

PGENETICS-D-23-00299R2 

A digital twin reproducing gene regulatory network dynamics of early *Ciona* embryos indicates robust buffers in the network 

Dear Dr Satou, 

We are pleased to inform you that your manuscript entitled "A digital twin reproducing gene regulatory network dynamics of early *Ciona* embryos indicates robust buffers in the network" has been formally accepted for publication in PLOS Genetics! Your manuscript is now with our production department and you will be notified of the publication date in due course.

With kind regards,

Livia Kovacs

PLOS Genetics

On behalf of:
